# Application of Soundproofing Materials for Noise Reduction in Dental CAD/CAM Milling Machines

**Eun-Sung Song** [1], **Young-Jun Lim** [2,*], **Joongsoo Lee** [3], **Jung-Bon Moon** [3] and **Bongju Kim** [1,*]

1   Clinical Translational Research Center for Dental Science, Seoul National University Dental Hospital, Seoul 03080, Korea; songeunsung@gmail.com
2   Department of Prosthodontics and Dental Research Institute, School of Dentistry, Seoul National University, Seoul 03080, Korea
3   DDS Inc., Gasan digital 1-40 2, Geumcheon-gu, Seoul 08591, Korea; jslee@aegisdds.com (J.L.); ceo@aegisdds.com (J.-B.M.)
*   Correspondence: limdds@snu.ac.kr (Y.-J.L.); bjkim016@gmail.com (B.K.); Tel.: +82-02-2072-4455 (B.K.); +82-02-2072-2940 (Y.-J.L.)

**Abstract:** Soundproofing materials are widely used in various fields as a passive measure to reduce noise. Despite this, there have been a few studies on the application of soundproofing materials on medical equipment, which is the main cause of noise in a medical environment. Despite the increasing number of studies on active noise control for the noise reduction of machines, it is difficult to apply customized noise control—i.e., specific control measures according to the various characteristics of that noise—due to its high cost and low effectiveness. Therefore, research on passive noise control using soundproofing materials is required for effective noise control. The 3D CAD/CAM milling machine, which is an essential device in the digitalized dental environment, is causing various problems as a new noise source. This study investigated the noise of the milling machine and considered its characteristics in application of an efficient soundproofing material for noise reduction. Additionally, a soundproofing material performance test was conducted to select an appropriate soundproofing material based on the noise characteristics of the milling machine. As milling machines cause noise issues in hospitals, the study results were analyzed in considering practical aspects for immediate application to actual sale products. This study suggests that the application of Thinsulator and a triple soundproofing mat (butyl 100% + aluminum + sound-insulating material) is effective in the noise reduction of milling machines.

**Keywords:** 3D CAD/CAM milling machine; passive noise control; soundproofing material

## 1. Introduction

The noise from machines such as handpieces in dental environments has become a cause of hearing loss in dentists and has evoked fear in patients [1,2]. Many different methods have been tried and tested to improve the satisfaction of patients and staff and generate a more pleasant environment at the hospital or clinic, and the reduction of unnecessary noise, among others, has been recognized and studied as an essential element [3–5]. Recently, various advanced medical machines have been introduced and concurrently recognized as new noise sources in dental environments. In particular, the emergence of three-dimensional computer-aided design/computer-aided manufacturing (3D CAD/CAM) has enabled the conversion of an existing manual process to a new digital process with the advantages of reduced cost and saved time, as a dental prosthesis can now be designed and manufactured in a continuous workflow [6]. However, despite having an effective manufacturing method, the milling machine is difficult to place within the dental environment, and noise control in the work environment

must still be implemented to suppress the excessive and unpleasant noise due to cutting that is still experienced even if it is placed in an alternative location, such as a separate room [7]. Thus, our study aims to analyze the noise characteristics of the 3D CAD/CAM milling machine, which causes a new type of noise in the dental environment, and to propose an effective noise reduction plan.

Noise reduction methods include both active and passive noise control. Active noise control is a technique that actively reduces noise by analyzing the wave form of the sound in real time and generating a control sound. Though it is effective for low frequencies, it has a high cost and is difficult to use in three-dimensional spaces [8]. On the other hand, passive noise control is a technique of passively reducing noise through a direct application to the noise source. It has been widely used due to its cost-effectiveness [9]. Therefore, in this paper, we present an effective noise control method considering the accurate identification of the characteristics of the noise source as well as the environment requiring noise control.

The fundamental solution to addressing the noise of machines in a medical environment is to remove the cause of noise during the manufacturing process while considering the usage environment. However, in the case of already commercialized products, precise analysis of the characteristics of the noise generated from a machine is needed. Consequently, it is critical to identify the noise transmission paths and the causes of such noise in a milling machine [10]. Our previous study suggested an active noise control method by identifying the noise characteristics of the milling machine but also reported the disadvantage of having to generate the control sound depending on where the machine is installed [11].

For the application of the passive noise reduction method, the basic noise reduction effects of the soundproofing and sound-insulating material that are applicable to the machine were analyzed. Due to the characteristics of the milling machine noise, in various work environments, the passive method of reducing noise by attaching noise-absorbing or sound-insulating materials to the pathways delivering noise from the noise source of the machine to the motor, housing, or bracket may be more effective than the active noise control method. However, it is critical to keep the use of soundproofing materials to a minimum by only using them in the optimal places, otherwise problems such as overheating may result. There have been an increasing number of studies on the noise-reducing effect of soundproofing and sound-insulating materials on medical machines and in the installation environment for the improvement of medical services and medical environments. However, there have been hardly any studies on dental environments [12–15]. Therefore, the purpose of this study is to present an effective noise reduction method using passive noise control through the application of soundproofing materials that is independent of the installation environment.

## 2. Research Methodology

To cut down the noise from milling machines, the causes of noise were first determined, and noise characteristics and control methods were subsequently analyzed and verified through experimentation. The process of reducing milling machine noise was classified into three phases: noise investigation, verification of causes for the noise, and the selection of a noise control method. The investigation method and procedure are presented in Figure 1.

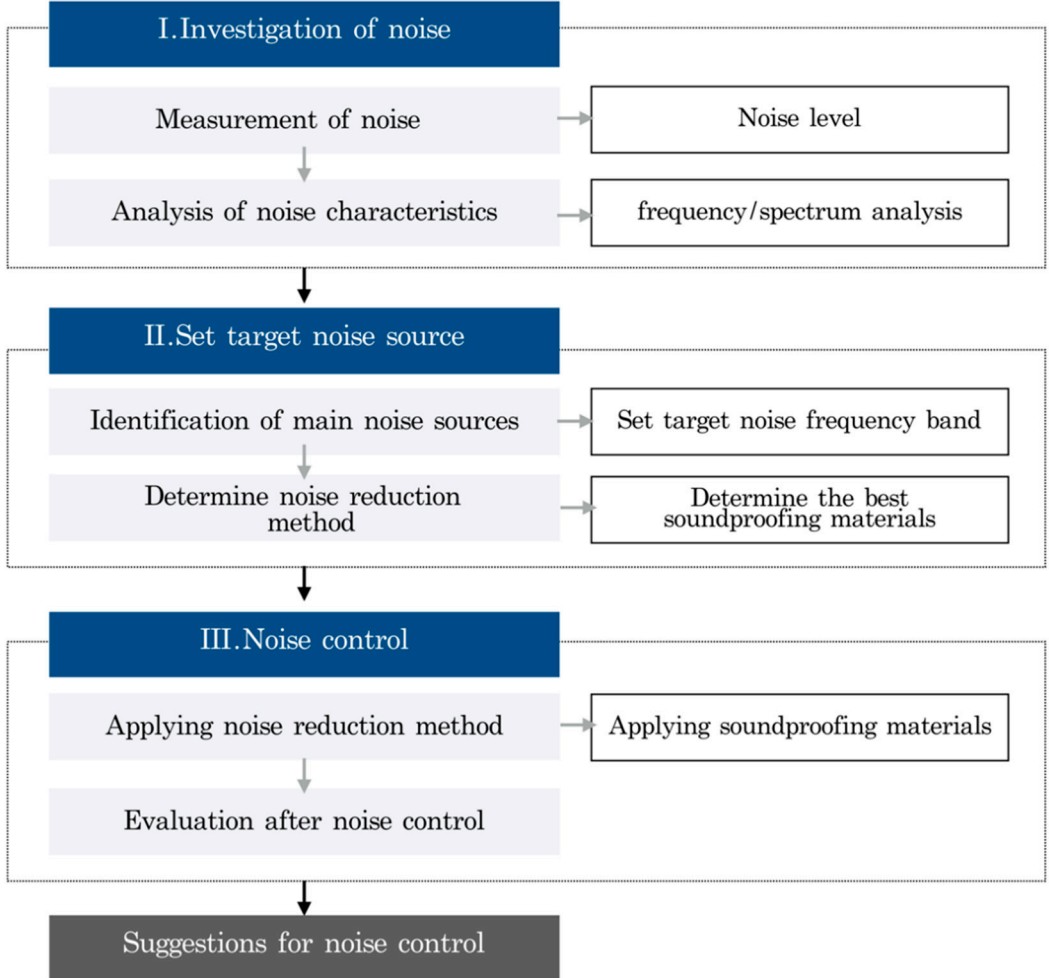

**Figure 1.** The noise control method process involving the application of soundproofing material to the milling machine.

*2.1. Investigation of Noise*

2.1.1. Noise Measurement

An Aegis HM, DDS (South Korea), was used to measure the noise of a milling machine, and zirconia (e max CAD 14) was used as the material (Figures 2 and 3A). The noise measurements were conducted using a measurement microphone (EMM-6 electret measurement microphone, Dayton Audio, Springboro, OH, USA) and the noise level meter (Lutron Sl-4013, Lutron, Taipei, Taiwan) (see Figure 3B). In order to identify each factor of the milling machine that generates noise, the housing (case) was removed, as shown in Figure 2, and the internal noise was additionally measured. Measuring the internal noise without the case was done to investigate the main spot where noise was generated. For the identification of the noise source in the milling machine, the external noise was first measured, followed by measurement of the internal noise. A dental milling machine is equipped with multi-axis high-speed milling, a drilling tool, and a rotating cutter and is controlled by a computer. The milling machine used in this research consists of a touch screen, spindle motor, and servomotor in the front and a control computer, air pump, and water pump in the back, as presented in Figure 2. This also has an internal frame and a case. The control computer, air pump, and water pump in the back are considered factors that might cause the noise. Therefore, the rear part was additionally measured to seek the causes of noise released outside the milling machine and the locations during internal noise measurement. The measurement points and times selected are shown in Figure 3C. The highest

noise level was selected by measuring the noise level at two or more appropriate measuring points. The noises were measured $5 \times 30$ s each within 20 min of the cutting processing time, and their average was recorded as the average noise. All experiments were independently repeated three times. The noise meter microphone was placed on the tripod that was installed at the measuring point, and the noise was measured while the target noise source was under normal operation.

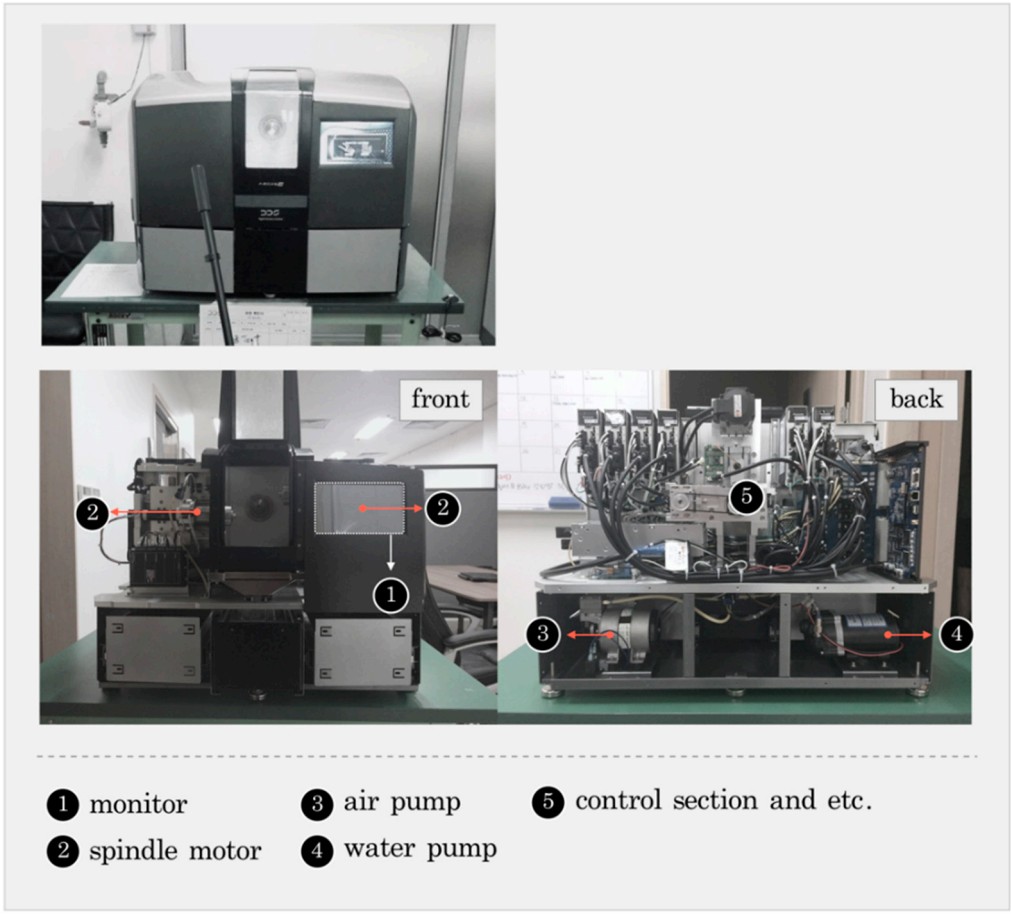

**Figure 2.** Features of the dental milling machine and the internal structure.

Figure 4 summarizes the noise level of the milling machine. The sound absorption level during the external noise measurement of the milling machine had a maximum deviation of 2.5 dB, ranging from 64.5 to 67 dB, and the sound absorption level during the internal noise measurement of the milling machine had a maximum deviation of 1.5 dB, ranging from 68.7 to 70 dB. However, the front of the milling machine had the highest sound absorption level, followed by the back, during both the internal and the external noise measurements.

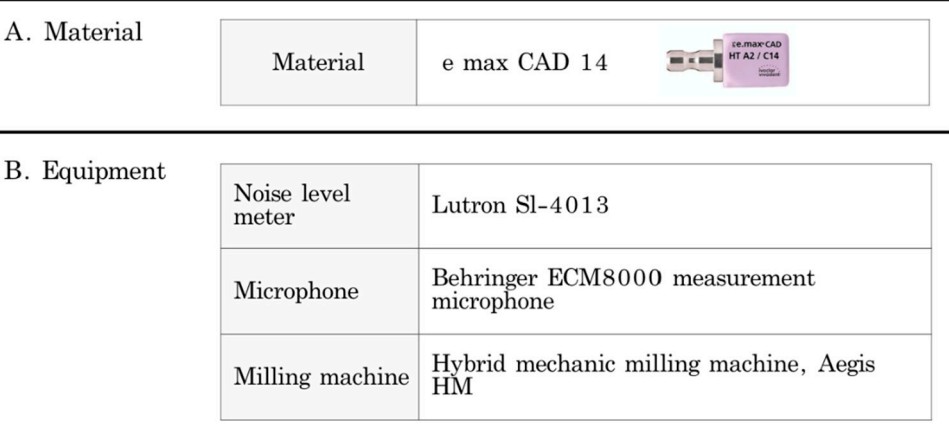

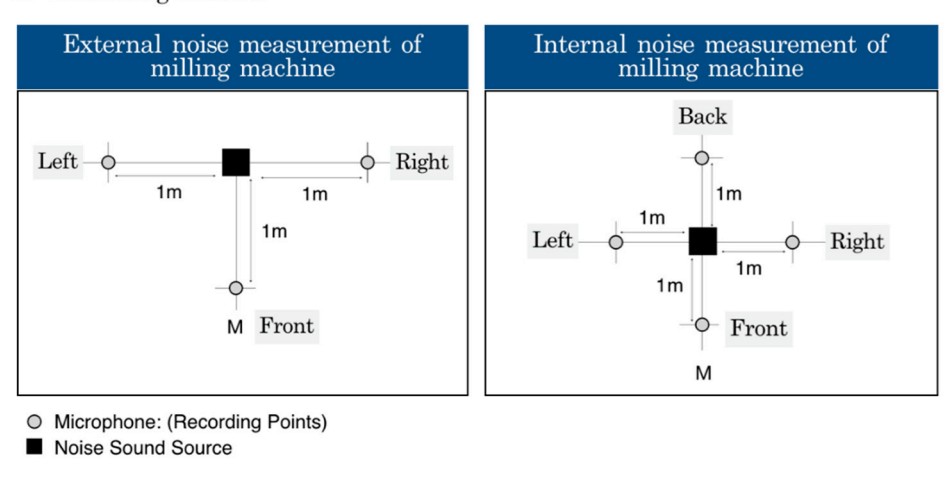

**Figure 3.** Experimental scheme. (**A**) material; (**B**) equipment; (**C**) measurement method.

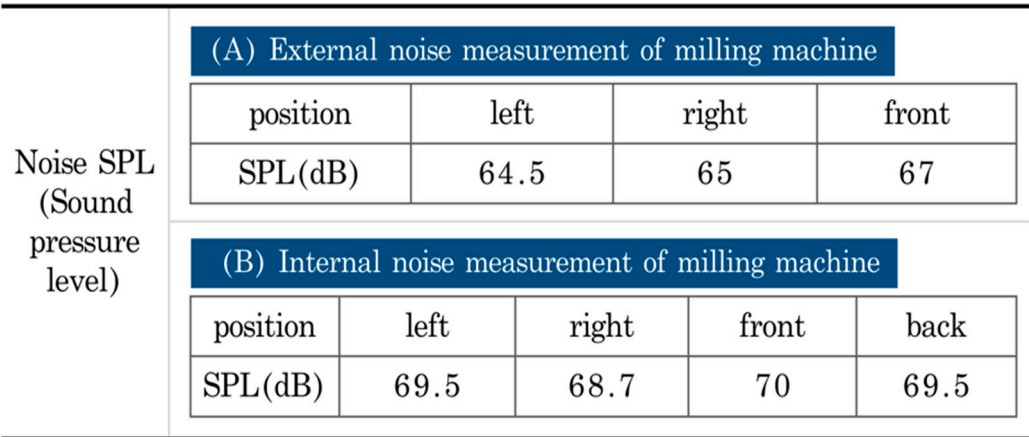

**Figure 4.** Noise level of the milling machine (SPL = sound pressure level). (**A**) external noise; (**B**) internal noise.

### 2.1.2. Analysis of Noise Characteristics

To analyze the noise characteristics of a milling machine, FFT (fast Fourier transform) analysis and spectrogram analysis were employed, focusing on frequency ranges. The program used was Adobe

Audition CC 2015. FFT analysis was used to analyze the frequencies of generated noise and the size of the frequency components. In addition, spectrogram analysis was executed to observe changes in the frequency domain and sound levels. This helps in understanding the overall frequency range of all noise at a glance. The analysis results are indicated in Figure 5. In the graph of Figure 5 showing each result, the X axis and the Y axis of the FFT analysis refer to the frequency (Hz) and the sound level (amplitude, dB), respectively. The X and Y axes of the spectrogram analysis respectively refer to time and frequency (Hz). Spectrogram analysis displays expanded ranges of frequency on a log scale, and the FFT analysis also uses a log scale to help in displaying all frequency components at once. The window size used for FFT analysis in this study is 4098, and the window type is Hamming.

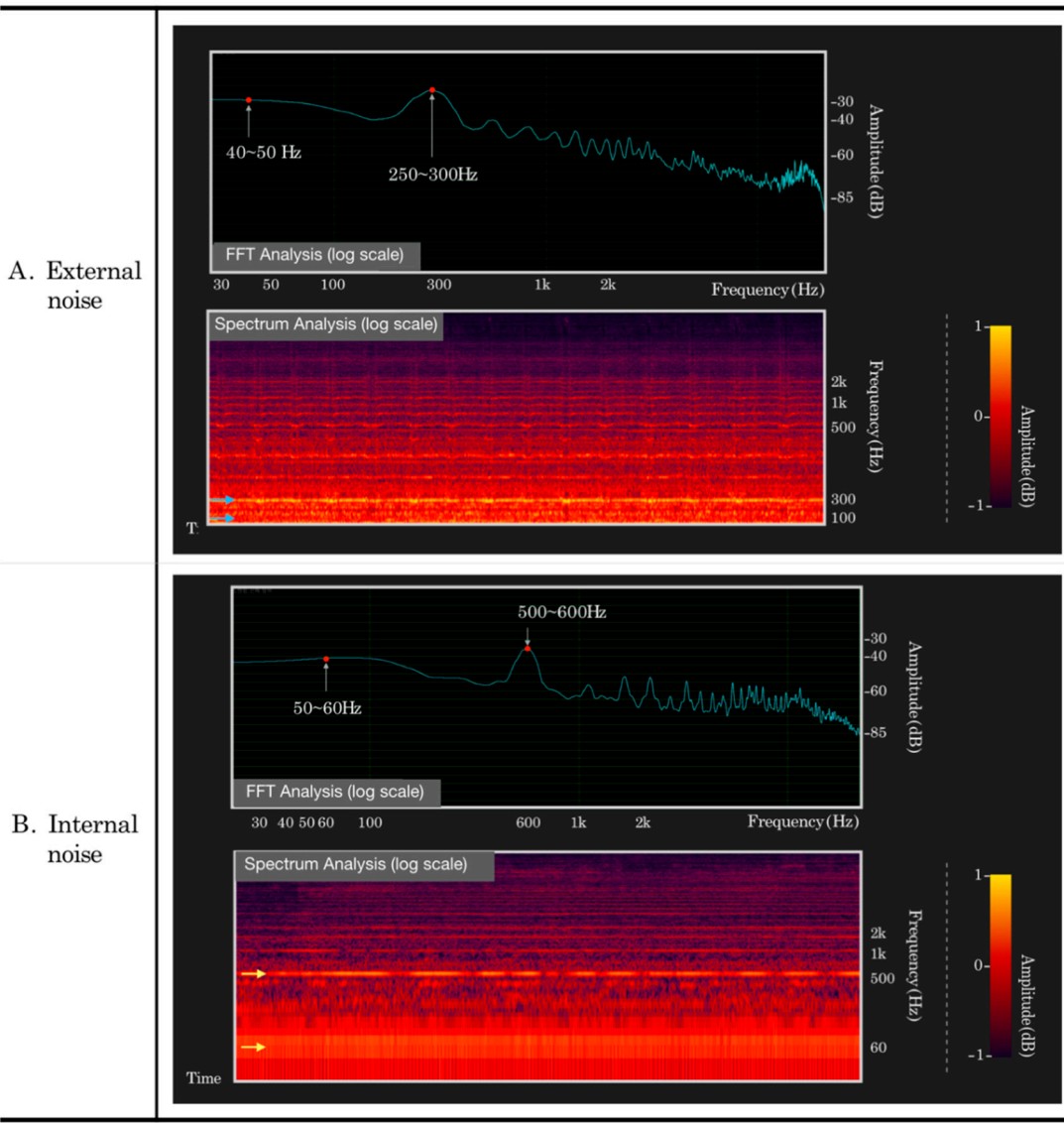

**Figure 5.** Analysis of noise source at the front of the milling machine. (**A**) external noise; (**B**) internal noise.

As shown in Figure 3 and Section 2.1.1, the internal noise was measured after measurement of the external noise (Figure 5). The noise generated from the milling machine is released in many different directions, so it is structurally difficult to control the noise and its source [16]. In order to identify the noise source and to develop feasible noise control plans, the noise released to the outside was measured, and the internal structure was then examined. An additional investigation of the internal

noise was conducted. The loudest noise released to the outside from the housing ranged between 250 and 300 Hz, followed by the second loudest noise ranging between 50 and 60 Hz (Figure 5A). The result of measuring the internal noise showed that the loudest noise frequency ranged between 500 and 600 Hz, followed by the second loudest noise frequency ranging between 40 and 50 Hz (Figure 5B). It was found that in spite of the shift in the peak frequency band where the loudest noise is generated, the main peak frequency was generated in two spots. The result of comparing the measurement of the internal noise without housing with the measurement of the noise released to the outside showed that the noise level was reduced and that the peak frequency band shifted due to the presence of the external housing, as shown in Figure 4.

## 3. Target Noise Source Setting and Prevention Technology Selection

### 3.1. Identifying and Analyzing the Cause of Noise

When observing a frequency spectrum of milling machine noise, there is a peak frequency which can explain the noise characteristics, although this may depend on the manufacturing company [11]. To apply effective noise control in this study, the peak frequency was discovered, and the frequency features of the milling machine noise were examined on the basis of the peak frequency.

Figure 4 shows that the strongest noise is generated at the front and back of a milling machine. The milling machine applied in this study comprises a spindle motor in the front and two pumps in the back, as shown in Figure 2. The spindle motor units (for cutting zirconia after bur installation) in the front and the two pumps in the back were chosen for noise analysis. The measurement method is shown in Figure 6A. The analysis result in Figure 6 shows the two spots (front: the motor with a spindle; back: the pump) that were used to analyze the frequency source, whereas the graph shows the frequency and the noise level of these two spots.

In order to identify the cause of the noise, it was confirmed that the peak frequency bands matched each other from the results of Figures 5B and 6B. The frequency of 560 Hz matched that of the 500–600 Hz bands responsible for generating the loudest noise during the measurement of noise released to the outside, and the frequency of 60 Hz matched that of the 50–60 Hz bands that generated the second loudest noise. Therefore, it is considered that these two spots are the main sources of the milling machine noise.

### 3.2. Setting the Target Noise Source and Determining the Noise Reduction Method

The selected target noise spot is where the strongest level of noise occurred, and the peak frequency range of generated noise refers to the target frequency range. The target noise was set based on the range of frequencies corresponding to the strongest level of noise in Figures 5 and 6. The method for controlling noise may vary depending on a variety of conditions of the noise, such as the frequency characteristics of the noise sources, regular or irregular noise sources, and repetition [17]. Thus, it is critical to establish what the most effective target noise source is, and apply the most appropriate method that is fitting for the noise source. In this study, the target noise source was established in consideration of the noise control effect depending on the noise controllability and human hearing characteristics. Human ears are most sensitive to frequencies between about 500 Hz and 6 kHz, and are less sensitive to frequencies above and below these (see Figure 7B). Brooks et al. [18] reported that an active noise control method can be effectively applied to reducing noise below the 500 Hz frequency band (Figure 7A). Thus, we concluded that the reduction of the target noise source at 560 Hz is more effective than the other target noise source of 60 Hz. Moreover, passive noise control using the difference in soundproofing performance is considered effective for the milling machine with the 560 Hz band as the peak frequency. Following the analysis result, the selected method of noise reduction was attachment of the soundproofing material to the inside of the housing. The most appropriate soundproofing material was selected based on the test results for noise-absorbing performance centering on the target frequency band of 560 Hz for each soundproofing material (Table 1).

A. Measurement method

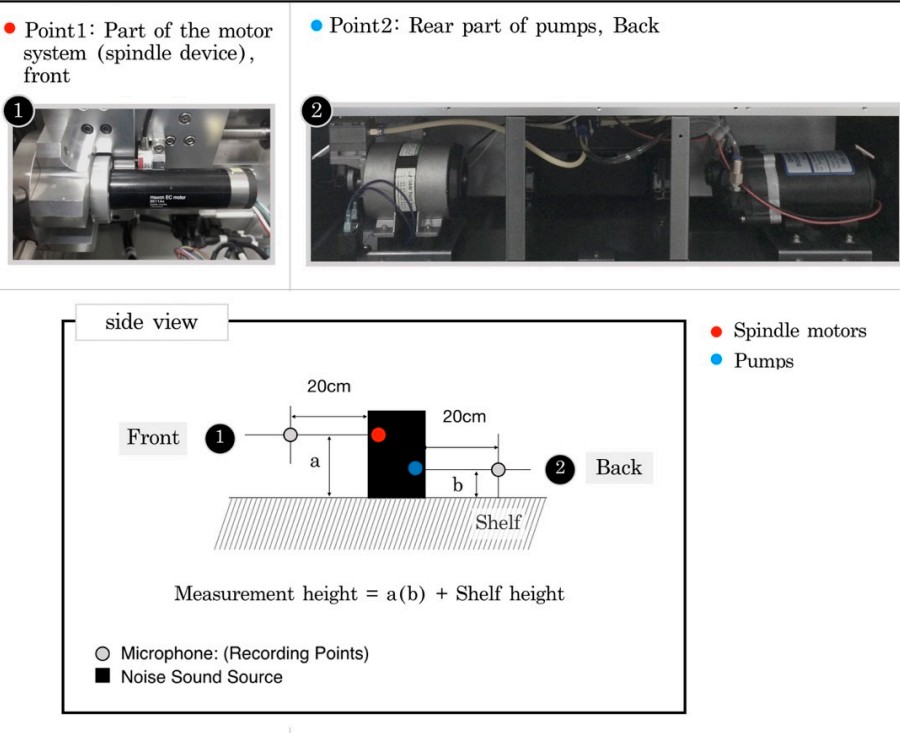

B. Results of Spindle Motors and pumps measurement to identify and analyze the main peak frequency sources

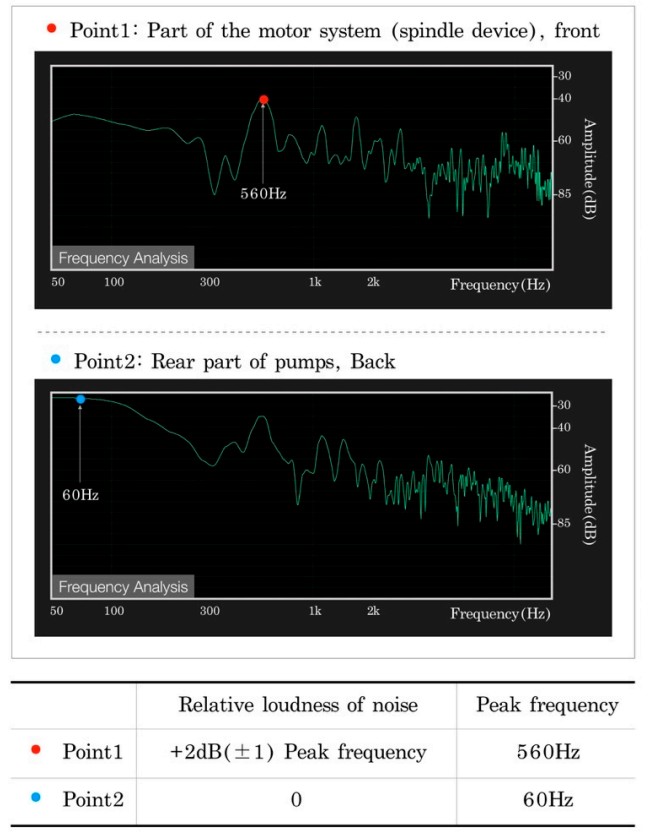

|  | Relative loudness of noise | Peak frequency |
|---|---|---|
| ● Point1 | +2dB(±1) Peak frequency | 560Hz |
| ● Point2 | 0 | 60Hz |

**Figure 6.** Spindle motor and pump measurements to identify and analyze peak frequency. (**A**) measurement method; (**B**) results of the spindle motor and pump measurement to identify and analyze the main peak frequency.

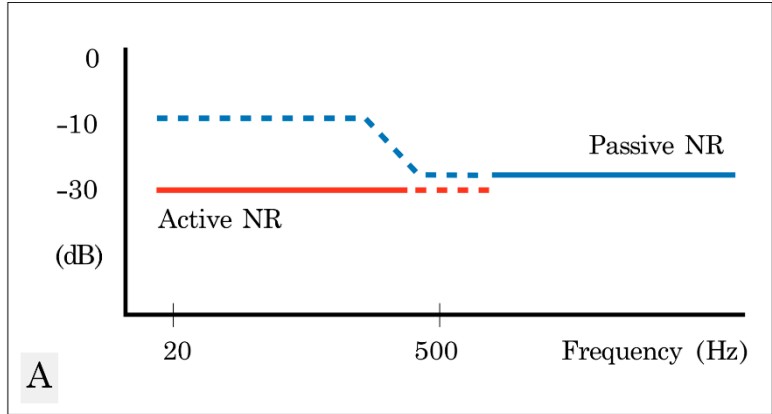

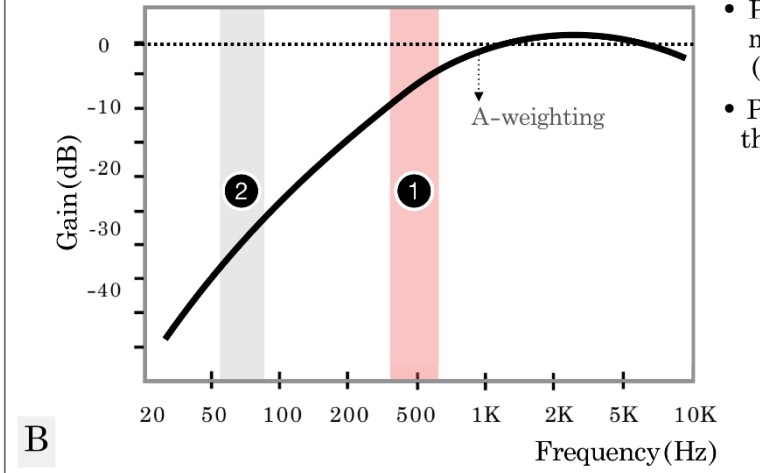

**Figure 7.** Target noise source setting and prevention technology selection. (**A**) operational range of active and passive noise reduction (active NR and passive NR); (**B**) frequency weighting curves. A-weighting is commonly used for a rough estimate of the frequency sensitivity of human hearing.

**Table 1.** Performance evaluation results of various materials.

| | Type | 1. Target Frequency Noise Level (SPL (dB) at 560 Hz) | 2. Recorded Milling Machine Noise Level (SPL (dB), 10 s on Average) | Thickness (mm) |
|---|---|---|---|---|
| A | Artboard | 64.1 | 75.4 | 9 |
| B | Rubber noise insulation | 57.0 | 76.4 | 2 |
| C | Sponge | 76.6 | 78.9 | 5 |
| D | Aluminum | 78.9 | 82.5 | 2 |
| E | Sponge | 72.8 | 79.0 | 5 |
| F | Thinsulator | 68.2 | 76.8 | 10 |
| G | Triple soundproofing mat (Butyl + Aluminum + Sound-insulating material) | 58.2 | 72.1 | 10 |

## 4. Determining the Best Soundproofing Materials

### 4.1. Performance Evaluation of Soundproofing Materials

A soundproofing material performance test was conducted to select the most suitable soundproofing material according to the frequency characteristics of milling machine noise. Thicker material and higher frequency sound increase sound absorption efficiency. Moreover, inserting an air layer increases sound absorption efficiency [19–21]. For evaluating the performance of soundproofing materials, the experiment was focused on three conditions: the scope of soundproof frequency, the thickness, and the applicability of each soundproofing material.

Since soundproofing materials do not absorb sound at all audible frequencies, pure tones with a target frequency of 560 Hz were generated, and the recorded noise of the milling machine was then played to measure the noise levels and the noise reduction effects of the individual soundproofing materials (see Figure 8). For the purpose of securing the basic data to be used as the selection criteria for the sound-absorbing/-insulating materials, the noise reduction performance was evaluated for seven types of various sound-absorbing/-insulating products (an artboard, rubber sound insulation, a sound-absorbing sponge, aluminum, a sponge, Thinsulator, and a triple soundproofing mat (butyl + aluminum + sound-insulating material)) (see Figure 9). Firstly, the sound-absorbing/-insulating materials were selected according to the characteristics of the products, and the sound-absorbing/-insulating characteristics of various materials were analyzed against the noise from the milling machine. The measurement method is described in Figure 8A. As shown in Figure 8B, a sample (2.5 m × 4 m) case with a size of 1400 mm × 700 mm × 700 mm was manufactured. Sound sources were installed in one corner of the case, and these were measured using a microphone (Dayton Audio EMM-6 electret measurement microphone) and a noise level meter (Lutron Sl-4013) in the opposite spot to record the changes of the soundproofing materials. The change in noise was then recorded for each material. All experiments were independently repeated three times. It is normal to calculate a sound absorption coefficient for each frequency (1/3 octave) to evaluate the sound absorption efficiency and sound absorption performance of soundproofing materials inside a reverberation chamber. This study, however, aimed to observe changes in noise released outside the milling machine, and not to compare the soundproof characteristics of soundproofing materials. Therefore, we monitored the actual changes in the noise from the milling machine in general environments such as a hospital, and not in a reverberation chamber.

In order to obtain basic data, a performance evaluation of varying materials was performed for their noise-absorbing effect. Seven types of soundproofing materials—an artboard, rubber sound insulation, a sound-absorbing sponge, aluminum, a sponge, a Thinsulator, and a triple soundproofing mat (butyl + aluminum + sound-insulating material)—were employed, as shown in Figure 5. The result of the noise reduction level for each of the seven types of soundproofing material is shown in Table 1. The soundproofing material with the lowest overall noise level was found to be the triple soundproofing mat (butyl + aluminum + sound-insulating material), followed by artboard, rubber sound insulation, and Thinsulator, in that order. However, among them, it was found that the artboard was difficult to attach to the curved inner side of the machine due to its hardness which also represents a difficulty in terms of commercial or sale purposes. Thus, it was excluded from further consideration.

## A. Scheme of performance test for soundproofing materials

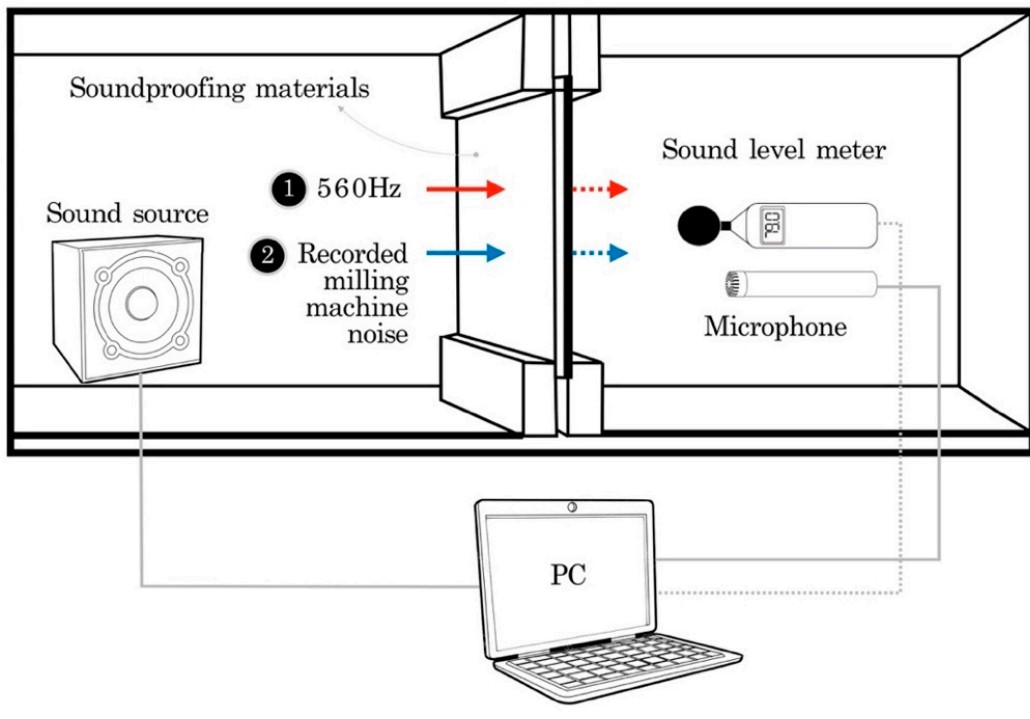

## B. Custom-made test box structure

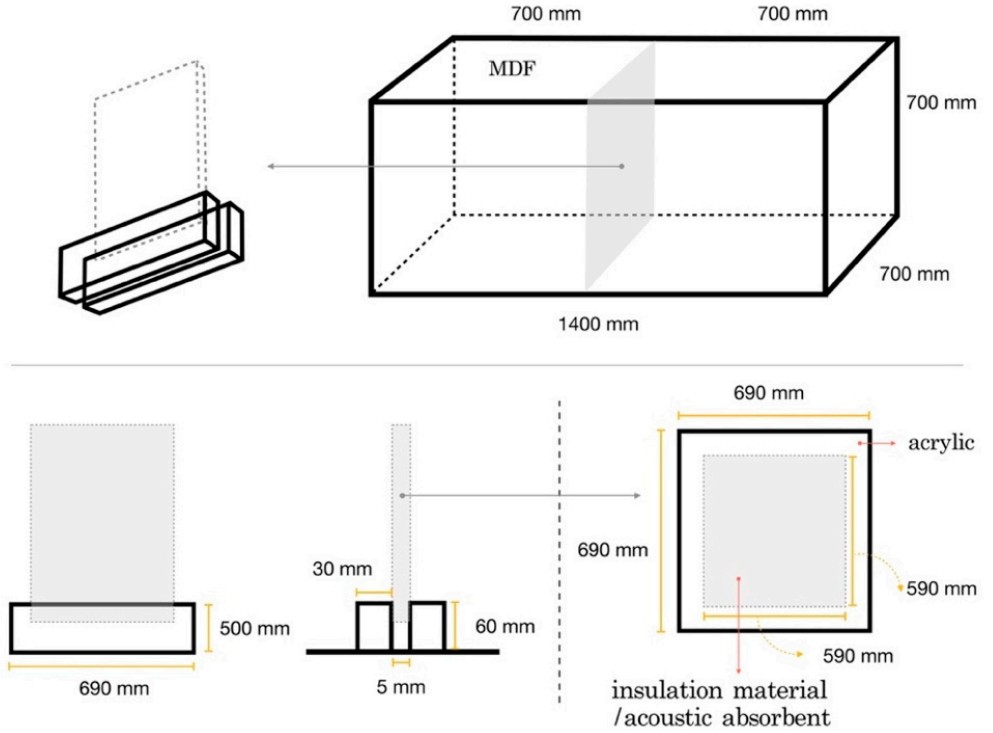

**Figure 8.** Method for evaluating the performance of soundproofing materials. (**A**) scheme of the performance test for soundproofing; (**B**) a custom-made test box structure.

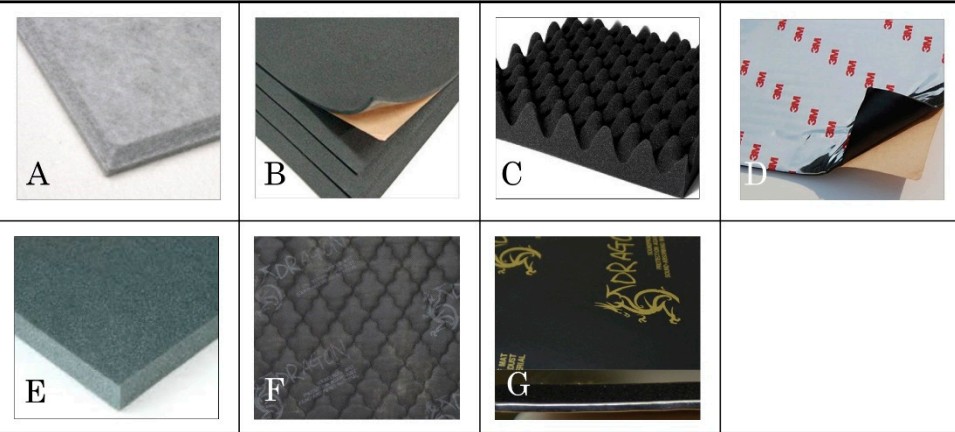

**Figure 9.** Types of soundproofing materials. (**A**) artboard; (**B**) rubber sound insulation; (**C**) sound-absorbing sponge; (**D**) aluminum; (**E**) sponge; (**F**) Thinsulator; (**G**) triple soundproofing mat (butyl + aluminum + sound-insulating material).

*4.2. Determining the Best Soundproofing Materials*

The noise reduction level was measured following the application of soundproofing materials (Thinsulator and the triple soundproofing mat (butyl + aluminum + sound-insulating material)) attached to the milling machine, as shown in Figure 10, as they were selected as the most appropriate of the seven types of soundproofing material. As shown in Figure 10A, the noise was measured 50 cm away from the noise source to the left, right, and front directions. The soundproofing material was attached to the milling machine housing for evaluation, as shown in Figure 10B. All experiments were independently repeated three times. The triple soundproofing mat (butyl + aluminum + noise insulating material) is well known to be effective in noise insulation, dust proofing, and dust elimination [22]. Thinsulator, a representative sound-absorbing material, was also combined to enhance target noise reduction [23]. Figure 11 shows the results of attaching and applying the selected or combined (to enhance the noise reduction efficiency) soundproofing materials directly to the milling. The combination of Thinsulator and the triple-soundproofing mat showed the greatest noise reduction effect. When only the triple-soundproofing mat was used, the noise reduction was found to be greatest around 560 Hz of the target frequency. On the other hand, when the combination of Thinsulator and the triple-soundproofing mat was used for reducing the overall noise, the noise was reduced across all frequencies rather than only in a certain frequency range, as shown in the graph.

As shown in Table 2, the results of applying Thinsulator and the triple-soundproofing mat indicate that the target frequency noise level was reduced by 9.0 dB on average, while the total noise level was reduced by 8.3 dB. Based on the results shown in Figure 11 and Table 2, it is considered that the use of a triple-soundproofing mat and Thinsulator (sound-absorbing material) would be the most effective for soundproofing.

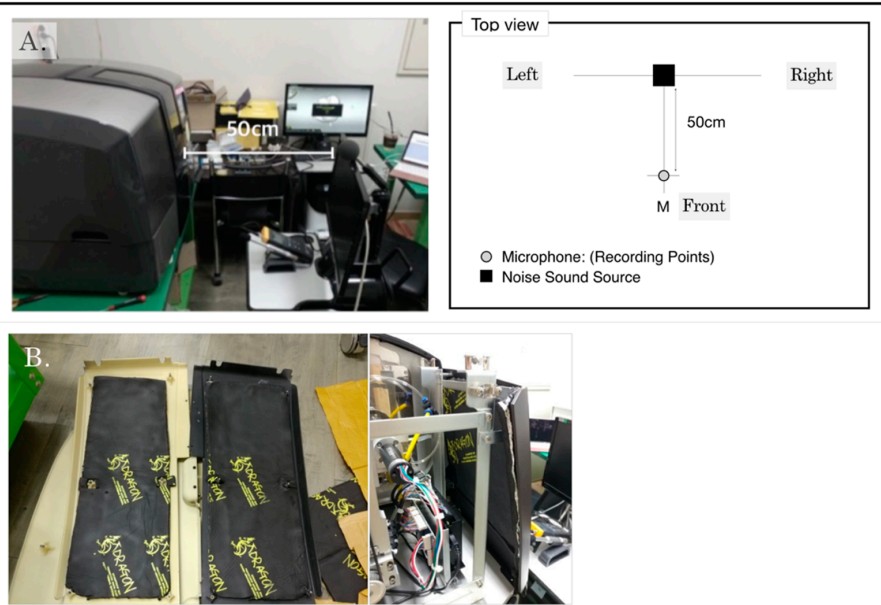

**Figure 10.** Measuring noise reduction after the application of soundproofing material. (**A**) measurement method; (**B**) the application of soundproofing material to the milling machine housing.

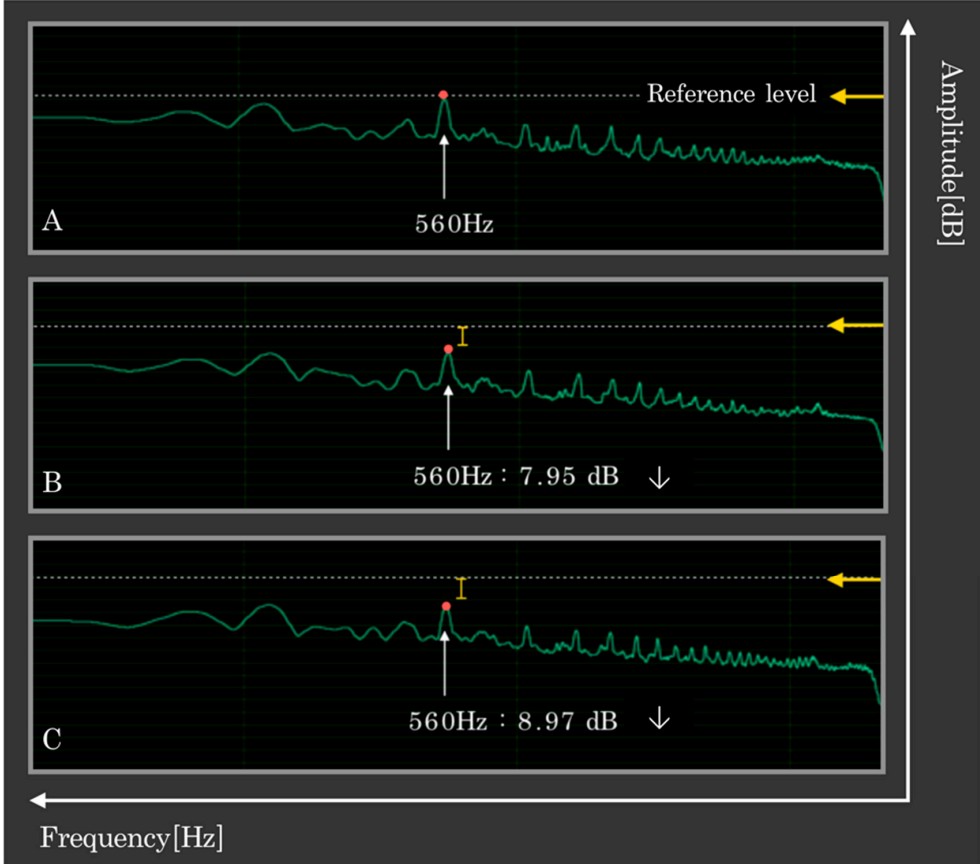

**Figure 11.** Application of passive noise control. (**A**) without soundproofing material; (**B**) with the triple-soundproofing mat; (**C**) with the Thinsulator and the triple-soundproofing mat.

**Table 2.** Summarized results of the noise reduction level.

| | Without Soundproofing Material | Triple-Soundproofing Mat | Thinsulator + Triple-Soundproofing Mat |
|---|---|---|---|
| Average noise level (dB) | 78.7 | 73.8 | 70.4 |
| Milling machine noise level (SPL) (dB, 10 s on average) | – | 5 | 8.3 |
| Target frequency noise level at 560 Hz (SPL), (dB, 10 s on average) | – | 8 | 9 |

## 5. Application and Results of the Noise Reduction Method

A previous study reported that the sound absorption rate of soundproofing materials differs according to the state of attachment [19–21]. It also depends on the conditions of the backside of the material (the presence/absence of a back wall of the material, the characteristics of the wall, the presence of an air layer between the material and the wall, the thickness of the air layer, etc.) Although the soundproofing material is thin, sound absorption is most effective when it is installed with some intervening space with respect to the wall. In order to soundproof low-pitched sounds, the soundproofing materials are installed at some distance from the wall, which usually reduces room space. However, as there is no free space in existing milling machines, the sound-absorbing material and -insulating material should be attached to the path through which noise is transmitted, with the minimum thickness that can be applied to milling machines. The combination of Thinsulator and a triple-soundproofing mat (butyl + aluminum + noise insulating material) was selected as most appropriate, as described in Section 4.2. The combination of Thinsulator and the triple-soundproofing mat (butyl + aluminum + noise insulating material) was attached to the milling machine housing for final evaluation, as shown in Figure 12. In the final evaluation, the temperature, the material thickness, and the attachable method were considered and applied in the test.

In the case of Thinsulator, the volume decreased by up to about 3 mm when it was compressed to the maximum. After applying 10 mm of the triple-soundproofing mat, the final thickness increased by approximately 1.3–1.5 cm (see Figure 10A,B). Figure 2 implies that it is impossible to apply this to all sides of the milling machine housing. For this reason, the areas for which attachment to the housing was not possible because of, e.g., a monitor, were excluded, as indicated in Figure 12C. Moreover, wide use of soundproofing materials may cause problems of heat generation due to the characteristics of milling machines. Therefore, it is necessary to consider the attachment space along with the heat and the final weight. After applying this, the weight increased by approximately 4 kg, and when it was measured under the same conditions as Figure 10, the noise decreased by 7.9 dB, from 78.5 to 70.6 dB. Regarding the heat problem, the temperature was independently measured and was observed to be maintained without any significant change. The final results after application are shown in Table 3.

**Table 3.** Final selection and application of sound-absorbing and -insulating material.

| | Without Soundproofing Material | After Applying Soundproofing Material |
|---|---|---|
| Average noise level (dB) | 78.5 | 70.6 |
| Temperature (°C) | 45 ± 2 | 45 ± 2 |

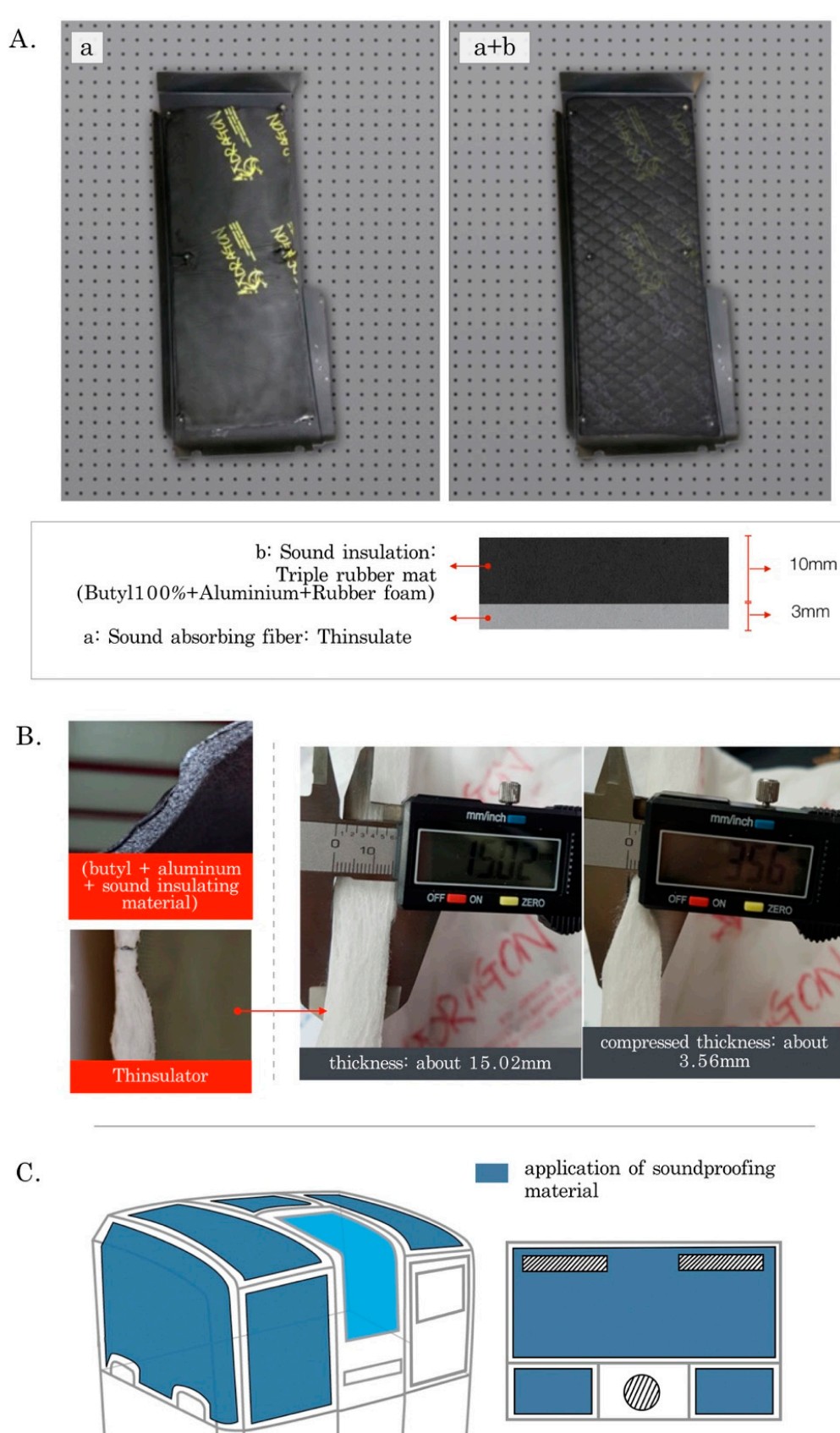

**Figure 12.** Optimal application of soundproofing materials to a milling machine. (**A**) the combination of Thinsulator and the triple-soundproofing; (**B**) thickness of each material; (**C**) the scheme of application.

## 6. Discussion and Conclusions

The milling machine noise that is released to the outside is approximately 65–70 dB when measured at a 1 m distance which, under continuous exposure and considering that normal conversations occur at a volume of 60 dB, is considered inappropriate for a pleasant environment [24,25]. Following the measurement of the internal/external noise of the milling machine, the target frequency band was determined to be 560 Hz in consideration of the peak frequency of the milling machine and human hearing sensitivity.

The noise control method involving the attachment of soundproofing material was selected for application to the milling machine structure and marketable products by taking into account the weight of the milling machine product, the attachment area, and other practical factors. Various types of soundproofing materials were tested for their noise-proofing performances. Furthermore, the sound absorption efficiency of soundproofing material varies depending on the setting, including the presence of a wall, the thickness of air layer, and the material attachment [19–21]. Therefore, it is critical that the soundproofing material is sparingly used in optimal spots rather than over large areas to minimize overheating, space usage, and weight. We carefully attached the housing to the motor area, which was the source of the loudest noise, as shown in Figure 12.

When the noise level of the machine was measured, the temperature was the same with or without the application of soundproofing material. Previous studies on passive noise control through soundproofing material have mainly focused on evaluation of the absorption rate or of sound-absorbing performances. However, such materials are commonly used in construction equipment and are seldom applied in commercial equipment, with vehicles being a notable exception. Furthermore, such materials have never been applied to medical equipment to address noise problems in a medical or dental environment. Technical developments and the emergence of new devices have led to the shift of technology to medical services in the medical or dental environment. It is necessary to create a more pleasant environment for the satisfaction of patients and staff, and this entails a reduction in noise from the machines used in such environments. Noise is an issue that has still not been satisfactorily resolved. Sound is not visible but directional, and it is hard to identify. In addition, it is more difficult to establish an objective standard for a pleasant environment, thus requiring more delicate attention, since sound is a sense that may affect our emotions and nervous system. As it is difficult to measure and evaluate noise based on the existing uniform standards, more detailed regulations, measurement methods, and standards for each device, particularly in medical and dental environments, are needed. This aspect must be further taken into consideration in future studies. Our results were obtained considering the material's practicality and immediate applicability to existing commercial dental milling machines that have been recognized as noise sources when installed in dental environments.

Our study suggests that, in consideration of the characteristics of noise from milling machines, the combination of Thinsulator and a triple-soundproofing mat (butyl 100% + aluminum + sound-insulating material) is more effective in noise reduction than any other tested soundproofing strategy.

**Author Contributions:** Conceptualization: E.-S.S.; methodology: Y.-J.L. and J.-B.M.; software: E.-S.S. and J.L.; validation: E.-S.S. and B.K.; data curation: E.-S.S. and B.K.; writing—original draft preparation: E.-S.S.; writing—review and editing: E.-S.S., B.K., and Y.-J.L.; visualization: E.-S.S., J.-B.M., and J.L.; supervision: B.K. and Y.-J.L. All authors have read and agreed to the published version of the manuscript.

**Funding:** This research was supported by a grant from the Korea Health Technology R&D Project through the Korea Health Industry Development Institute (KHIDI), the Ministry of Health and Welfare, Republic of Korea (grant number: HI15C1535), and the Technology Innovation Program (or Industrial Strategic Technology Development Program-Industrial Core Technology Development Project), funded by the Ministry of Trade, Industry and Energy (Grant number: 10062635).

**Conflicts of Interest:** The authors declare no conflicts of interest.

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
