# Peer review of "Application of Soundproofing Materials for Noise Reduction in Dental CAD/CAM Milling Machines"

_applsci, doi:10.3390/app10082768_

Round 1

Reviewer 1 Report

This paper use different noise cancelling material and measure the noise in practical experiments. However, the poor English and grammar make it very difficult to follow. The problems are throughout the paper but for example in the abstract lines 23 and 24 it says ‘In this study, the basic data was researched on the soundproofing materials made with various materials as an effective noise reduction method, in consideration of the milling machine’s noise characteristic.’ I assume this means the research was into the effectiveness of various soundproofing materials but it took several readings to understand what was being said.

Another problem is inconsistencies in the paper. For example it states in the abstract there has been few studies of soundproofing in a medical everonment but in the introduction it says there has been an increasing number of studies? Similarly the abstract says that soundproofing can be achieved by installing the machine in a separate space but I could not see any reference to the location of the machine in the paper?

Finally I found the conclusion quite confusing and I’m not entirely sure what the researchers conclusions and recommendations are. I cannot find any innovation in this research paper. 

Author Response

Dear Reviewer

Please confirm attached file.

Sincerely yours,

Reviewer 2 Report

General comments and recommendations:

The topic addressed in the article is of technical interest, as the acoustic comfort in healthcare facilities in which noisy equipment is present is a critical issue. The choice of the object of investigation, i.e. the noise reduction of CAD/CAM milling machines for dental applications, is also original.

The paper, however, needs a substantial revision to be accepted for publication (see the comments below). I also recommend a professional linguistic check, as the quality of the English in the article is very variable. Section 5. Discussion & conclusion, in particular, should be thoroughly revised.

Specific comments and recommendations

At the beginning of the article, a detailed description of the machine under investigation would be helpful, particularly for readers who are not familiar with this type of equipment. The description should identify the main elements of the machine mentioned in the paper (e.g. spindle unit, pumps, etc.), with reference to suitable figures (e.g. the pictures of Fig. 1 and Fig. 6).

The meaning of “External noise” and “Internal noise” should be better explained. Is internal/external related to the casing or body of the apparatus? Alternatively, is it related to separate parts of the apparatus? The schemes and pictures of Fig. 1 are quite difficult to interpret and do not answer my question.

At some point in the article (e.g. in the Introduction or in section 2.1.2 Performance evaluation of soundproofing materials) I suggest adding a discussion on the interplay between sound absorption and sound transmission in passive noise control strategies.

The terminology used in describing the acoustic measurements is often unprecise. Concepts such as “sound” and “noise” are used loosely and sometimes not appropriately. My recommendation is to use, in general, the terms “sound” or “acoustic” with reference to the sources and to the measured quantities, and the term “noise” with reference to the unwanted effects (e.g. annoyance). I noticed that the terms “sound pressure” and “sound pressure level”, which are by far the most common terms used in acoustics, are never mentioned in the paper.

in Section 2.1.1, I recommend to insert a table or figure summarizing the results of the noise measurement tests.

In section 2.1.2 Performance evaluation of soundproofing materials, the measurement procedure should be described in more detail. I also recommend to insert a table or figure summarizing the results of the materials evaluation tests.

Lines 143-145: the terms “left” (line 143), “back” (line 144) and “right” (line 145) should be referred to a clear graphical representation of the machine, which is not present in the draft article.

Lines 145-151: how were the noise sources identified? By placing a microphone close to the source or by other methods?

Line 153 and caption of Figures 6: please explain the meaning of the term “problem frequency source”.

Line 166: What do you mean with “regulation or repetition”?

Table 1:

  • The caption of the Table 1 “Sound absorption and sound insulation measurement” is confusing. I suggest using a title more closely related to the numerical results given in the table.
  • Sound pressure levels in dB should be rounded to 1 decimal place (this applies throughout the article).
  • Please provide in the text a precise definition of “Total noise” and “Target noise”.

Lines 208-209: What do you mean by “total noise level”? Is it the overall noise level over the entire audible spectrum?

Table 2: please round off the figures in dB to one decimal place.

Author Response

(The authors gave the same response as above.)

Round 2

Reviewer 1 Report

The English and grammar is improved but there is another concerned. 

Conclusion should summary your finding on your research. Some of your paragraph in conclusion should be in introduction as they are background such as some references. 

Author Response

Dear Reviewer

Reviewer#1

Point: Conclusion should summary your finding on your research. Some of your paragraph in conclusion should be in introduction as they are background such as some references.

Response: Thank you for your advice. We agree your opinion.

We changed the section name.

Line 321: 6. Discussion & conclusion

Sincerely,

Reviewer 2 Report

The paper has been significantly improved and all my comments have been adequately addressed. I just recommend this chance at Line 274: “the target noise frequency was reduced by 8.97 dB” -> “the target frequency noise level was reduced by 9.0 dB”

Author Response

Dear Reviewer

Reviewer#2

Point: The paper has been significantly improved and all my comments have been adequately addressed. I just recommend this chance at Line 274: “the target noise frequency was reduced by 8.97 dB” -> “the target frequency noise level was reduced by 9.0 dB”

Response: Thank you for your advice.

By following your suggestions, we changed the sentence.

Line 269: the target frequency noise level was reduced by 9.0 dB

Sincerely,